# FROM ATTACK TO RESTORATION: A TWO-STAGE DIFFUSION FRAMEWORK FOR FACE PRIVACY

## ABSTRACT

The surge of facial photos on social media has made unauthorized face recognition (FR) a serious threat to personal privacy. Existing diffusion-based privacy methods are vulnerable to the purification effect, which weakens adversarial signals, and their single-stage optimization struggles to balance deceptiveness and visual quality. To address this, we propose a two-stage face privacy protection framework. During the Identity Diversion stage, we introduce Negative Prompt Inversion (NPI) into the diffusion reverse process and incorporate an angular margin constraint to steer features toward a target identity in feature space—counteracting the dilution of adversarial signals at the source and mitigating gradient conflicts and trade-off issues. During the Visual Fidelity Restoration stage focuses on perceptual quality, using perceptual loss and regularization strategies to enhance naturalness while preserving the method's ability to deceive recognizers. Extensive experiments on the CelebA-HQ and LADN public datasets show that our approach achieves state-of-the-art protection success rates (PSR) while maintaining high image quality, underscoring its promise for privacy protection and real-world deployment.The code is available in the Supplementary Material.

## 1 INTRODUCTION

With the rapid advances of deep neural networks, face recognition (FR) has become widely used in identity authentication, mobile payments, and public security(Hill, 2022). However, the massive sharing of facial images on social media(Besmer & Richter Lipford, 2010) (Smith et al., 2012)and in public settings enables unauthorized FR systems to track user behavior, analyze social relationships, and even commit identity theft without consent, posing serious threats to personal privacy and security. Therefore, an urgent challenge is how to effectively defend against unauthorized face recognition while preserving the visual quality of images.

Existing research has explored various adversarial approaches to facial privacy protection. Early methods based on noise perturbations(Cherepanova et al., 2021) (Shan et al., 2020)(Yang et al., 2021)or adversarial patches(Komkov & Petiushko, 2021) (Xiao et al., 2021)can reduce recognition success to some extent, but they often introduce noticeable visual artifacts that degrade naturalness. Subsequently, makeup-style transfer methods (Hu et al., 2022)(Shamshad et al., 2023) (Sun et al., 2024)attempt to embed adversarial information as cosmetic effects on the face, thereby achieving privacy protection while maintaining a natural appearance. However, these methods typically rely on reference images or textual prompts, making fine-grained control difficult, and their cross-model transferability under black-box settings still needs improvement.

In recent years, the powerful generative and editing capabilities of diffusion models(Sohl-Dickstein et al., 2015) have opened new avenues for facial privacy protection. A representative work, DiffProtect(Liu et al., 2023), was the first to incorporate diffusion models into adversarial face protection by modifying semantic codes in the latent space to impersonate a specified target identity. However, during the reverse diffusion process it is affected by the so-called "diffusion purification effect"—the model tends to remove high-frequency adversarial perturbations during denoising—resulting in a drop in PSR. At the same time, excessive modification of the semantic code can introduce perceptible distortions to facial structure. To alleviate this issue, subsequent research(Salar et al., 2025) introduced learnable unconditional embeddings(Mokady et al., 2023) in the diffusion latent space to strengthen the retention of adversarial information. Nevertheless, because the method relies on

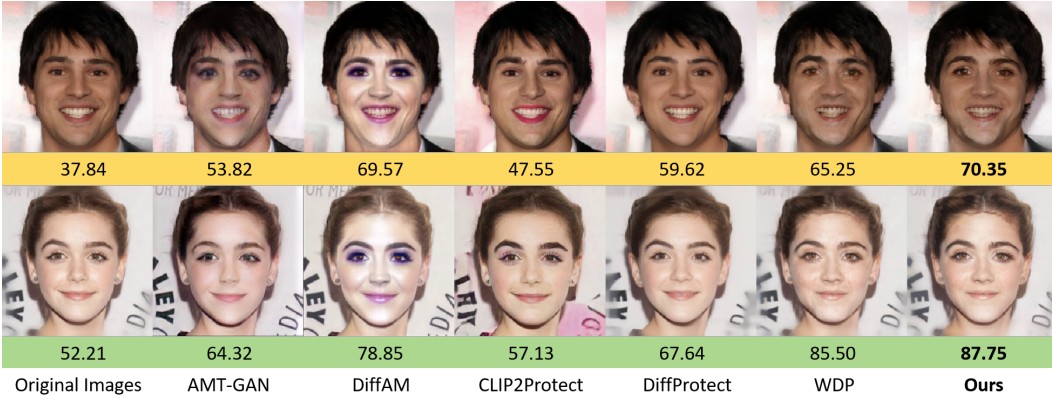

Figure 1: All the images above were generated using various facial privacy protection methods; the number beneath each image corresponds to the verification confidence score from the Face++ API.

indirect alignment via unconditional embeddings, it struggles to eliminate biases injected by the original content.Meanwhile, existing diffusion-based methods must trade off between image quality and PSR, so the achievable PSR is constrained by the requirements on perceptual quality.

Accordingly, diffusion-based approaches(Liu et al., 2023)(Salar et al., 2025)to facial privacy protection have gradually demonstrated a superior trade-off between visual quality and black-box(Dong et al., 2018) transferability compared with traditional noise perturbation and makeup-transfer strategies. Inspired by this, and under the same task setting as prior work, we propose a two-stage scheme: directly applying adversarial optimization in the latent space while introducing structure-preservation constraints, so as to maintain high-quality, natural appearance while effectively defending against various unauthorized face recognition models.

Specifically, we build a two-stage diffusion-based adversarial optimization framework. During the Identity Diversion stage, the model follows the reverse diffusion trajectory to generate adversarial samples and uses target identity features as supervision, maximizing the misclassification probability in feature space and thereby substantially enhancing deception capability. During the Visual Fidelity Restoration stage, we further minimize perceptual loss within the latent space and introduce a dynamic rollback mechanism, which restores fine-grained textures and improves visual naturalness without sacrificing attack success. This design breaks the conventional "single-pass diffusion with a single objective" paradigm, enabling the protected images to combine strong misleading power against FR systems with high perceptual quality.In conclusion, our contributions are threefold:

(i) **Negative-Prompt Inversion as a de-biased initializer.** We replace step-wise null-text optimization with Negative-Prompt Inversion (NPI), providing a de-biased and steady starting point for inversion. This mitigates early gradient oscillations induced by semantic bias and yields faster, more stable convergence to the target identity.

(ii) **Decoupled two-stage objective.** We separate adversarial identity diversion from visual fidelity restoration into two dedicated stages. This removes objective interference present in single-stage designs, stabilizes training, and improves the end-to-end balance between deception success (PSR) and visual quality.

(iii) **Comprehensive evaluation under black-box settings.** We conduct broad comparisons on public benchmarks against strong baselines under black-box conditions (Dong et al., 2018). Our method achieves substantially higher PSR while preserving high visual quality and remaining competitive on perceptual metrics, highlighting its practical potential for facial privacy protection.

## 2 RELATED WORK

**Noise-based methods.** Noise-based privacy protection perturbs the input face at the pixel level so that the perturbed image's embedding in a recognition model either moves toward a target identity (impersonation) or away from the source identity. Representative transfer-based attacks include

PGD(Madry et al., 2017), MI-FGSM(Dong et al., 2018), and TI-DIM(Dong et al., 2019), which enhance black-box transferability via iterative updates, momentum, and translation invariance, respectively; TIP-IM(Dong et al., 2019) further designs an update rule tailored for targeted identity protection. Although such methods can achieve nontrivial protection under black-box settings, they essentially overlay "non-semantic" noise masks across the entire image, often leaving perceptible artifacts—e.g., noticeable grain and spurious textures—that undermine user experience and facial naturalness. In practice, this artifact–effectiveness trade-off is a core limitation of noise-based approaches and has motivated subsequent work to reorganize adversarial evidence into more semantic, localized edits, thereby improving visual plausibility without sacrificing attack success.

**Patch-based methods.** Patch-based approaches conceal identity by overlaying visually salient adversarial patches on limited facial regions, thereby disrupting feature extraction in FR models. Early works design wearable adversarial accessories, such as the colorful eyeglass frames of Adv-Glasses(Sharif et al., 2019) and the adversarial brim of Adv-Hat(Komkov & Petiushko, 2021), which can be physically worn to mislead FR systems in real-world settings. To improve black-box transferability, (Xiao et al., 2021) further construct digital adversarial patches and regularize them on a low-dimensional data manifold represented by generative models. However, the localized/restricted editing region of patches introduces an inherent trade-off: because perturbations are confined to small and often conspicuous areas, their ability to fully suppress FR accuracy is limited, and they tend to reduce the naturalness of the protected images.

**Makeup-based methods.** Makeup-based defenses embed adversarial cues as cosmetic edits. Early GAN methods such as Adv-Makeup(Yin et al., 2021) and AMT-GAN(Hu et al., 2022) transfer makeup from a reference image but often introduce artifacts and spill into non-makeup regions. CLIP2Protect(Shamshad et al., 2023) shifts editing to the StyleGAN latent space with CLIP guidance(Li et al., 2021), removing the need for reference images yet offering only coarse control. DiffAM(Sun et al., 2024) adopts a text-guided diffusion pipeline that first removes and then synthesizes fine-grained adversarial makeup, markedly improving naturalness and black-box attack success. However, all these methods still require retraining or re-tuning when the target identity changes, limiting practical deployment.

**Diffusion-based methods.** As state-of-the-art probabilistic generative models, diffusion models can produce highly realistic, ultra–high-resolution images. Studies show this capability can also be used to craft adversarial examples: for example, DiffProtect(Liu et al., 2023) protects faces by modifying the semantic latent codes of Diff-AE(Preechakul et al., 2022), but such direct edits disrupt facial structure and consistency with the original image. Another line of work(Salar et al., 2025) (For convenience, hereafter referred to as WDP.)operates in the latent space and introduces null-text optimization(Mokady et al., 2023) to attenuate the diffusion process's purification effect, yet relying on this alone cannot fully suppress it, leaving PSR limited. More critically, most methods must balance image quality and PSR simultaneously, thereby constraining overall performance and scalability.

## 3 METHOD

### 3.1 OVERVIEW

We propose the first two-stage, diffusion-based framework for facial privacy protection (as shown in Fig.2). In the Identity Diversion stage, we jointly employ Negative Prompt Inversion (NPI) and an angular-margin loss to rapidly generate intermediate images that strongly mislead recognition models in feature space, thereby boosting adversarial deceptiveness. We then move to the Visual Fidelity Restoration stage: guided by the LPIPS(Snell et al., 2017) perceptual loss and further constrained by latent-space L2 regularization with a dynamic rollback mechanism, we restore high-frequency details and fine textures without weakening the attack established in Stage I, substantially improving naturalness and perceptual quality.

This "deceive first, then restore" strategy breaks the inherent trade-off between protection success rate (PSR) and image quality that plagues single-stage methods. Experiments on CelebA-HQ(Karras et al., 2017) and LADN(Gu et al., 2019) show that our approach not only achieves a significantly higher PSR than mainstream single-stage baselines, but also maintains visual quality metrics such

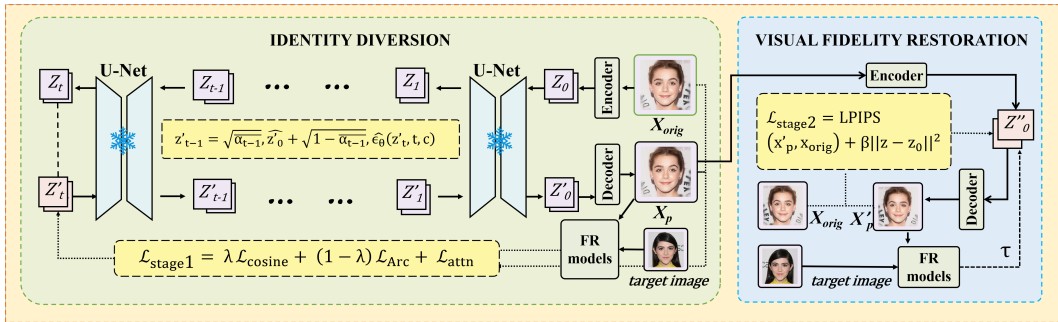

Figure 2: Overview of the two-stage framework. Our method employs a two-stage strategy for image protection:Identity Diversion takes the original face $x_{\text{orig}}$ as input and, guided by NPI and the angular-margin loss, drives the sample in feature space toward the target image, yielding a highly deceptive intermediate $x_p$; Visual Fidelity Restoration then starts from $x_p$, refines details using LPIPS and a latent-space $L2$ regularizer, and applies a threshold-controlled dynamic rollback to obtain a higher-quality $x_p'$ without weakening the attack.

as FID(Heusel et al., 2017), demonstrating practical potential for real-world deployment and establishing a new two-stage diffusion paradigm for facial privacy protection.

## 3.2 NEGATIVE PROMPT INVERSION

Unlike DDPM (Ho et al., 2020) and DDIM (Song et al., 2020), which carry out diffusion and denoising directly in pixel space—an approach that often introduces local blurring, texture degradation, and difficulties in maintaining long-term structural consistency—Stable Diffusion (Rombach et al., 2022a) moves the diffusion process into latent space, markedly reducing computational and memory overhead while further enhancing the stability and controllability of high-resolution image generation.

Because both diffusion and denoising occur in latent space, subsequent semantic guidance and loss constraints can be imposed directly on this compact representation. However, when we wish to edit a real face $x_{\text{orig}}$, the image must first be "brought back" onto the model's latent trajectory to provide a valid starting point. Accordingly, we employ DDIM inversion to map $x_{\text{orig}}$ into the latent sequence $(z_1, z_2, \ldots, z_t)$.

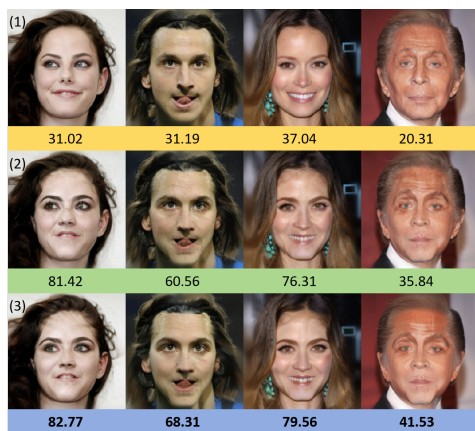

Figure 3: Comparing NPI(Ours) vs purification from null-text suppression diffusion.Rows: (1) original image, (2) null-text, (3) NPI.

This constructs a one-to-one trajectory on the diffusion model's data manifold corresponding to the original image, allowing semantic guidance and loss constraints to be applied *in situ*. In doing so, we preserve reconstructability and facial-structure consistency while avoiding the distribution shifts and artifacts caused by indiscriminate perturbations. Because DDIM sampling can be regarded as the numerical solution of an ordinary differential equation, it furnishes an approximately invertible mapping along with a step-wise, controllable editing interface, enabling stable, high-fidelity image manipulation. The basic inversion procedure can be expressed as:

$$z_{t+1} = \sqrt{\frac{\bar{\alpha}_{t+1}}{\bar{\alpha}_t}}\, z_t \;+\; \sqrt{\bar{\alpha}_{t+1}}\left(\sqrt{\frac{1}{\bar{\alpha}_{t+1}} - 1} \;-\; \sqrt{\frac{1}{\bar{\alpha}_t} - 1}\right)\epsilon_\theta\big(z_t, t, \varnothing_t\big)$$

Figure 4: (Left) Top row: original image, single-stage result, two-stage result; bottom row: the corresponding difference maps with respect to the original. (Right) Illustration of how sample distributions change in feature space after incorporating the angular-margin loss.

Here, $\bar{\alpha}_t$ denotes the noise scaling factor, and $\epsilon_\theta(z_t, t, \varnothing_t)$ is the noise term predicted by the U-Net at timestep $t$. The DDIM sampling process is given as follows:

$$z_{t-1} = \sqrt{\frac{\bar{\alpha}_{t-1}}{\bar{\alpha}_t}} \, z_t \; + \; \sqrt{\bar{\alpha}_{t-1}} \left( \sqrt{\frac{1}{\bar{\alpha}_{t-1}} - 1} \; - \; \sqrt{\frac{1}{\bar{\alpha}_t} - 1} \right) \epsilon_\theta(z_t, t, \varnothing_t)$$

To mitigate the diffusion purification effect, WDP (Salar et al., 2025) adopts an indirect alignment based on the unconditional (null-text) embedding (Mokady et al., 2023), which requires repeatedly updating the negative-branch embedding at every diffusion step, thereby increasing computational and memory overhead. However, this mechanism struggles to remove biases injected by the original image content, limiting convergence and stability in the early inversion stage. Inspired by NPI (Miyake et al., 2025), we replace unconditional optimization at initialization with a fixed semantic embedding, directly constraining the inversion start point and attenuating content bias, so that the trajectory converges earlier and more stably toward the target while also improving the efficiency of adversarial sample generation.As shown in Fig.3, NPI outperforms the null-text(Mokady et al., 2023) optimization baseline. The core idea can be stated as:

$$\varnothing_t \approx C,$$

Here, $\varnothing_t$ denotes the diffusion model's unconditional embedding at step $t$, which we approximate with a constant semantic vector $C$. In this paper, we set $C = $ "face" to avoid semantic ambiguity introduced by complex descriptions and to emphasize face editing as the core objective.

### 3.3 STAGE I-IDENTITY DIVERSION

As illustrated in Fig. 2, we break the intrinsic trade-off between attack success rate and image quality that hampers single-stage pipelines (Liu et al., 2023; Salar et al., 2025) by introducing a two-stage optimization framework. The first stage—Identity Diversion—focuses exclusively on maximizing adversarial strength, laying a solid foundation for the subsequent high-fidelity restoration.

In striving for stronger adversarial performance, we observe that most existing face-privacy and attack methods (Hu et al., 2022; Liu et al., 2023; Sun et al., 2024; Salar et al., 2025) rely solely on a similarity loss to nudge the generator toward the target identity. Such a loss enforces only global directional alignment and fails to impose explicit inter-class separability in feature space, leaving the produced adversarial samples fragile when confronted with unseen recognition models. To address this weakness, we introduce an ArcFace(Deng et al., 2019)-style angular-margin loss: unlike the original ArcFace (Deng et al., 2019), which employs classification cross-entropy, our formulation directly imposes an angular margin between source and target features. The generated image must therefore not only approach the target in cosine space but also satisfy a tighter angular boundary, greatly enhancing its deceptive capability and cross-model generalization (see Fig. 4(Right)). The ArcFace-style loss used in this work is defined as:

$$L_{\text{Arc}} = \sum_{i=1}^{N} s \, \max\big(0, \, \cos\theta_i - \cos(\theta_i + m)\big), \qquad \cos\theta_i = \left\langle \frac{\mathbf{e}_{\text{src}}^{(i)}}{\|\mathbf{e}_{\text{src}}^{(i)}\|}, \frac{\mathbf{e}_{\text{tgt}}^{(i)}}{\|\mathbf{e}_{\text{tgt}}^{(i)}\|} \right\rangle.$$

Here, $\theta_i$ is the angle between the $i$-th pair of source and target embeddings, $m$ is the angular-margin hyperparameter, $s$ is the scaling factor. The term $N$ represents the number of source–target feature pairs in one computation and is used to average the loss over all pairs.

| Method | CelebA-HQ | | | | LADN | | | | Average |
|---|---|---|---|---|---|---|---|---|---|
| | IRSE50 | IR152 | Facenet | MobileFace | IRSE50 | IR152 | Facenet | MobileFace | |
| clean | 7.29 | 3.80 | 1.08 | 12.68 | 2.71 | 3.61 | 0.60 | 5.11 | 4.61 |
| *Noise-based* | | | | | | | | | |
| PGD (2017) | 36.87 | 20.68 | 1.85 | 43.99 | 40.09 | 19.59 | 3.82 | 41.09 | 25.60 |
| MI-FGSM (2018) | 45.79 | 25.03 | 2.58 | 45.85 | 48.30 | 25.57 | 6.31 | 45.01 | 30.63 |
| TI-DIM (2019) | 63.46 | 36.17 | 15.30 | 57.12 | 56.36 | 34.18 | 22.11 | 48.30 | 41.64 |
| TIP-IM (2021) | 54.40 | 37.23 | 40.74 | 48.72 | 65.89 | 43.57 | 63.50 | 48.32 | 50.06 |
| *Makeup-based* | | | | | | | | | |
| Adv-Makeup (2021) | 21.95 | 9.48 | 1.37 | 22.00 | 29.64 | 10.93 | 0.97 | 22.33 | 14.72 |
| AMT-GAN (2022) | 76.96 | 35.13 | 16.62 | 50.71 | 90.12 | 32.13 | 13.23 | 73.92 | 52.84 |
| CLIP2Protect (2023) | 81.10 | 48.42 | 41.72 | 75.26 | 91.57 | 53.31 | 47.91 | 79.94 | 64.90 |
| DiffAM (2024) | **92.00** | 63.13 | 64.67 | 83.35 | 95.66 | 66.75 | 65.44 | 92.04 | 77.88 |
| *Diffusion-based* | | | | | | | | | |
| DiffProtect (2023) | 67.75 | 60.14 | 35.19 | 64.33 | 54.51 | 44.27 | 31.33 | 50.90 | 51.05 |
| WDP (2025) | 88.87 | 67.25 | 59.53 | **91.57** | 95.78 | 70.18 | 62.05 | **98.17** | 79.17 |
| **Ours** | 90.58 | **79.97** | **74.56** | **91.57** | **96.99** | **83.18** | **87.04** | 96.99 | **87.61** |

Table 1: Protection Success Rate (PSR, %) on CelebA-HQ and LADN under the black-box setting.The highest value in each column is typeset in bold; the second highest is underlined.

As shown in Fig.2, in the first stage, the remaining loss terms include The Cosine Loss, by minimizing the cosine distance between the two embeddings, ensures that the adversarial sample is closely aligned with the target identity in embedding-space direction, providing the most direct "move-toward-target" driving force. Meanwhile, the self-attention regularization term constrains the attention distribution in latent space, effectively preventing the generated image from suffering severe structural distortions that could arise from an excessive focus on adversarial strength.

$$
\begin{cases}
L_{\text{Cosine}} = 1 - \cos\big(F(x_p), F(x_t)\big), \\
L_{\text{attn}} = \big\|S(z_{\text{adv}}) - S(\bar{z}_t)\big\|_2^2, \\
\mathcal{L}_{\text{stage1}} = \lambda\,\mathcal{L}_{\text{Cosine}} + (1-\lambda)\,\mathcal{L}_{\text{Arc}} + \mathcal{L}_{\text{attn}}.
\end{cases}
$$

Here, $F(x_p)$ and $F(x_t)$ denote the feature vectors of the adversarial sample $x_p$ and the target image $x_t$, respectively. The coefficient $\lambda \in [0,1]$ is a balancing weight that trades off between the traditional cosine-similarity constraint and the strengthened angular-margin constraint. Moreover, $S(z_{\text{adv}})$ and $S(\bar{z}_t)$ represent, respectively, the perturbed and unperturbed self-attention maps of the diffusion model at timestep $t$.

### 3.4 STAGE II-VISUAL FIDELITY RESTORATION

As shown in Fig.2, the core objective of stage of the Visual Fidelity Restoration is to further improve the perceptual quality of the generated images while preserving—without diminishing—the adversarial efficacy already achieved in the Identity Diversion stage, thereby attaining higher visual fidelity. By adopting this divide-and-conquer design, each stage can concentrate on its own optimization goal while leaving headroom for the other, breaking through the traditional single-stage bottleneck between attack success rate and image quality and offering a clear, operational pathway to enhance both simultaneously.(As shown in Fig.4(Left))

In this stage, we feed the adversarial images produced in the Identity Diversion stage together with their corresponding original faces into a Stable Diffusion model and use its variational auto-encoder to map the adversarial images to latent-space representations. All subsequent optimization is performed purely in latent space, which not only greatly increases computational efficiency but also exploits the natural-image prior inherent in the diffusion model's latent manifold, ensuring realism in the reconstructed results. The process can be formalized as the following constrained optimization problem:

$$
\min_{\mathbf{z}}\ \mathcal{L}_{\text{stage2}}(\mathbf{x}'_p, \mathbf{x}_{\text{orig}}) \quad \text{s.t.} \quad \cos\text{-sim}(\mathbf{x}'_p, \mathbf{t}) \geq \tau,
$$

Here, $\mathbf{t}$ denotes the feature embedding of the target identity, and $\tau$ is the face-recognition threshold determined by a specified FAR. The primary objective is to minimize a proxy Fréchet Inception Distance(FID)(Heusel et al., 2017) to improve perceptual quality; concretely, we use Learned Perceptual Image Patch Similarity(LPIPS)(Zhang et al., 2018) as the perceptual loss. To prevent the

| Method | PSR ($\uparrow$) | FID ($\downarrow$) | PSNR ($\uparrow$) | SSIM ($\uparrow$) |
|---|---|---|---|---|
| TIP-IM(2021) | 50.06 | 38.7357 | 33.2089 | 0.9214 |
| Adv-makeup(2021) | 14.72 | **4.2282** | **34.5152** | **0.9850** |
| AMT-GAN(2022) | 52.84 | 34.4405 | 19.5045 | 0.7873 |
| DiffAM(2024) | 77.88 | 26.1015 | 20.5260 | 0.8861 |
| DiffProtect(2023) | 51.05 | 28.2912 | 24.2070 | 0.8785 |
| WDP(2025) | 79.17 | 18.0380 | 27.8664 | 0.8538 |
| **Ours** | **87.61** | 16.0342 | 27.1780 | 0.8512 |

Table 2: Quantitative comparison of privacy protection (PSR) and visual-quality metrics. The highest value in each column is typeset in bold; the second highest is underlined.

latent variables from drifting too far from their initial values, we further include an L2 regularization term in the loss. The overall objective can be written as

$$\mathcal{L}_{\text{stage2}} = \text{LPIPS}\big(x'_p,\ x_{\text{orig}}\big) + \beta \|\mathbf{z} - \mathbf{z}_0\|^2,$$

To ensure the attack remains effective during optimization, we introduce a threshold-guard mechanism: after each latent-space decoding, we compute the cosine similarity with the target embedding. If the similarity falls below the threshold $\tau$, the algorithm immediately rolls back to the most recent latent state verified as safe, thereby preventing the attack from failing.

# 4 EXPERIMENTS

## 4.1 EXPERIMENTAL SETUP

**Datasets.** Following the experimental protocol of AMT-GAN (Hu et al., 2022), we evaluate our method on two public datasets—CelebA-HQ (Karras et al., 2017) and LADN (Gu et al., 2019). CelebA-HQ(Karras et al., 2017) is a high-resolution version of CelebA containing 30 000 face images; consistent with AMT-GAN(Hu et al., 2022), we randomly sample 1 000 images, each from a distinct identity, for our experiments. LADN(Gu et al., 2019) offers 333 no-makeup and 302 makeup images, of which 332 no-makeup images are used as the test set. For both datasets, we further divide the data into four subsets, each linked to a specific target identity. All images within a subset are required to impersonate that identity, mirroring the four preset target identities used in AMT-GAN(Hu et al., 2022).

**Target Models.** To evaluate the defense effectiveness against face-recognition (FR) systems, we select four widely used black-box FR models: IRSE50(Hu et al., 2018), IR152(He et al., 2016), FaceNet(Schroff et al., 2015), and MobileFace(Chen et al., 2018). During fine-tuning, we randomly choose three of these models to provide gradient feedback, leaving the remaining one aside for strictly black-box testing. Following previous work(Shamshad et al., 2023), all input images are first aligned and cropped with MTCNN(Zhang et al., 2016) before being fed into the FR models.

**Baselines.** We benchmark our method against ten representative approaches covering three families: *(i) Pixel–noise attacks:* PGD(Madry et al., 2017), MI-FGSM(Dong et al., 2018), TI-DIM(Dong et al., 2019) , and the strongest pixel-noise baseline TIP-IM(Yang et al., 2021); *(ii) Digital-makeup attacks:* Adv-Makeup(Yin et al., 2021), AMT-GAN(Hu et al., 2022), CLIP2Protect(Shamshad et al., 2023), and the current makeup-based SOTA DiffAM(Sun et al., 2024); *(iii) Diffusion-based attacks:* DiffProtect(Liu et al., 2023) and the latest SOTA WDP(Salar et al., 2025).

**Metrics.** Consistent with CLIP2Protect(Shamshad et al., 2023), we use PSR to evaluate the effectiveness of the proposed method relative to baselines. PSR is computed with the false accept rate (FAR) set to $0.01$(Hu et al., 2022)(Shamshad et al., 2023). In addition, we evaluate the visual quality of protected images using Fréchet Inception Distance (FID)(Heusel et al., 2017), peak signal-to-noise ratio (PSNR, in dB), and structural similarity index (SSIM)(Wang et al., 2004).

**Implementation Details.** Our implementation is built on(Salar et al., 2025). The Identity Diversion stage fixes the text prompt to "face" and performs $T_1 = 35$ optimization steps during inversion; the loss weight is set to $\lambda = 0.5$. The Visual Fidelity Restoration stage II runs $T_2 = 70$ optimization

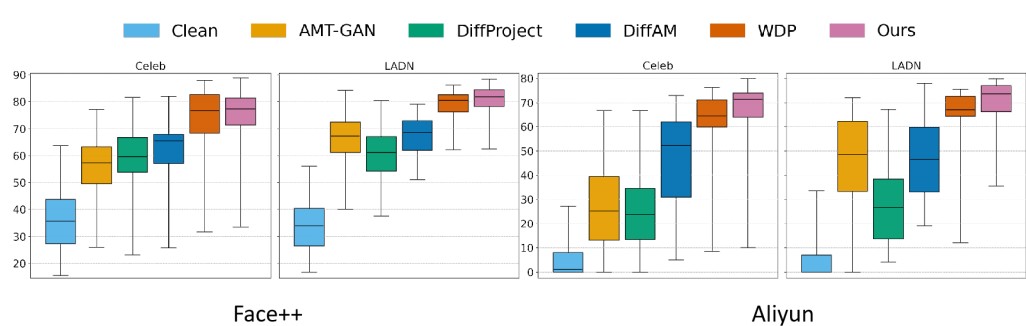

Figure 5: The confidence scores returned from commercial APIs, Face++ and Aliyun

steps ; the threshold $\tau$ follows the setting used in(Hu et al., 2022)(Salar et al., 2025). For more experiments, please refer to the appendixA.6.

## 4.2 COMPARATIVE EXPERIMENTS

In this section, we evaluate both our method and the baselines under black-box(Dong et al., 2018) conditions against four pre-trained face-recognition (FR) models, assessing their protective performance and the quality of the generated images.

**Personal Facial Privacy Protection Evaluation.** On the CelebA-HQ(Karras et al., 2017) and LADN(Gu et al., 2019) datasets, we adopt a black-box(Dong et al., 2018) evaluation protocol: each of four common FR models is taken in turn as the target model, while the remaining three serve as surrogate models to generate adversarial examples; for each target model, we evaluate four target identities and report the average. To better reflect real-world usage, the target face images used at test time come from the same subjects as in training but are different images.As shown in the Tab.1 with the threshold set to FAR = 0.01 and using PSR as the metric, our method shows a clear advantage over representative baselines—namely the noise-based TIP-IM(Dong et al., 2019), the makeup-based DiffAM(Sun et al., 2024), and WDP(Salar et al., 2025)—with average PSR improvements of approximately 37%, 9.7%, and 8.4%, respectively.

**Qualitative assessment of visual quality.** We evaluate the generated images from both quantitative and qualitative perspectives; the quantitative results are reported in the Tab2. Overall, because Adv-Makeup(Yin et al., 2021) applies only lightweight edits around the eyes, it achieves strong performance on visual-quality metrics; however, its conservative changes markedly limit PSR. By contrast, our method attains an advantage in FID(Heusel et al., 2017), indicating that the protected images are closer to the real distribution and appear more natural subjectively. On individual metrics such as SSIM and PSNR(Wang et al., 2004), we are on the same order as DiffProtect(Liu et al., 2023),DiffAM(Sun et al., 2024)and WDP(Salar et al., 2025), while maintaining stronger privacy protection. The qualitative visualizations likewise show that our edits are more localized with fewer artifacts, preserving overall structure and facial recognizability and yielding more robust privacy protection—thus achieving a better overall balance between naturalness and protection strength.

**Attack Performance on Commercial APIs.** In Fig.5, we evaluate our method and several baselines under conditions closer to real-world use: we employ two mainstream commercial face recognition (FR) APIs—Face++ and Aliyun—and randomly sample 100 images from each of the CelebA-HQ(Karras et al., 2017) and LADN(Gu et al., 2019) datasets for protection and evaluation. The confidence scores range from 0 to 100, with higher values indicating greater similarity between the protected image and the target identity. The results show that our approach achieves high confidence scores on both APIs; notably, WDP(Salar et al., 2025) also performs competitively on Face++.

## 4.3 ABLATION STUDY

As illustrated in Tab.3, the ablation study shows that each module contributes meaningfully to the overall pipeline: with all components enabled, we obtain a marked boost in PSR while still delivering high-quality, visually faithful images.

| Methods | | | Evaluation Metrics | | | |
|---|---|---|---|---|---|---|
| NPI | Angular-margin loss | Stage II | PSR (↑) | FID (↓) | PSNR (↑) | SSIM (↑) |
| × | × | × | 79.17 | 18.0380 | 27.8664 | 0.8538 |
| ✓ | × | × | 79.32 | 29.3841 | 25.6454 | 0.8119 |
| × | ✓ | × | 85.60 | 34.2472 | 25.1762 | 0.7924 |
| × | × | ✓ | 79.17 | 12.3357 | 28.9720 | 0.8788 |
| ✓ | ✓ | × | 87.61 | 31.8926 | 25.3224 | 0.8022 |
| ✓ | × | ✓ | 79.32 | 15.0564 | 27.6237 | 0.8577 |
| × | ✓ | ✓ | 85.60 | 16.6114 | 27.2330 | 0.8504 |
| ✓ | ✓ | ✓ | 87.61 | 16.0342 | 27.1780 | 0.8512 |

Table 3: Quantitative comparison of privacy-protection effectiveness and visual-quality metrics across ablation settings.

**NPI.** Introducing NPI(Miyake et al., 2025) approximates the unconditional embedding(Mokady et al., 2023) $\varnothing_t$ with a fixed semantic vector, giving DDIM inversion a stable, debiased starting point in diffusion space. This strategy markedly suppresses diffusion purification, concentrates gradient directions, and reduces early-stage oscillations, allowing the optimization trajectory to converge faster and more stably toward the target identity(as shown in Fig.3). As shown in Tab.3, when combined with the angular-margin loss, NPI(Miyake et al., 2025) boosts the PSR from 85.60% to 87.61% with virtually no additional computational overhead, underscoring its effectiveness.

**Angular-margin loss.** Introducing the angular-margin loss imposes a strict margin on the angle between source and target feature vectors in embedding space, effectively adding an "angular gate" beyond cosine similarity. This mechanism markedly enhances the deceptiveness of adversarial examples(As shown in Fig.4(Right)) and their cross-model generalization; even when applied in isolation it delivers a substantial boost in PSR.

**Stage II.** The two-stage framework—"deceive first, then restore"—substantially enhances perceptual quality while preserving attack strength. With Visual Fidelity Restoration stage in place, our method lowers the FID(Heusel et al., 2017) from 31.89 to 16.03 without any reduction in PSR. When the Visual Fidelity Restoration stage is added on top of WDP (Salar et al., 2025), FID(Heusel et al., 2017) likewise drops from 18.04 to 12.34, again with PSR unchanged. These results confirm that the two-stage design achieves a superior trade-off between visual quality and adversarial effectiveness compared with single-stage optimization, validating the "strengthen attack first, then refine quality" strategy.

As shown in Tab.3, although our images are not of the highest visual quality, our method achieves the best PSR. We prioritize PSR over purely perceptual metrics because, in black-box face-recognition settings, a protected image attains true "unrecognizability"—and thus real security value—only when its PSR is sufficiently high.

## 5 CONCLUSIONS

This paper proposes a diffusion-based two-stage framework that decouples attack enhancement from visual restoration. In the Identity Diversion stage, the method rapidly guides samples toward the target identity in feature space, thereby counteracting the dilution of adversarial signals caused by diffusion denoising. Subsequently, in the Visual Fidelity Restoration stage, it further refines image details and textures while preserving the established deception. At present, the loss function in the Visual Fidelity Restoration stage is primarily aimed at improving FID(Heusel et al., 2017); future work may introduce multi-objective losses that jointly optimize FID(Heusel et al., 2017), SSIM, and PSNR(Wang et al., 2004), or adopt alternative architectural designs, to enhance visual quality across multiple dimensions. The two stages are modular and compatible with standard inversion pipelines and guidance schedules, enabling straightforward integration across different datasets and face recognition (FR) backbones.

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

## A APPENDIX

To facilitate a deeper understanding of our work and provide comprehensive experimental details, we include an appendix with several supplementary sections. These sections elaborate on key aspects that complement the main text, including the application of large language models, a detailed background on latent diffusion models, the selection of target images, the face recognition models used in our evaluation, specific parameter settings, and additional visualization results. This supplementary material aims to ensure transparency and reproducibility while offering readers further insights into the methodologies and experiments discussed in the main paper.

### A.1 THE USE OF LARGE LANGUAGE MODELS

During the writing of this paper, we leveraged large pre-trained language models (LLMs) to rapidly retrieve and synthesize literature highly relevant to our study, ensuring both comprehensive and up-to-date coverage. We also used these models to polish the draft's language and provide accurate Chinese-English translation, standardizing terminology and streamlining the narrative so that the manuscript more closely adheres to academic conventions.

### A.2 LATENT DIFFUSION MODEL

**Latent-space diffusion.** Conventional diffusion models (e.g., DDPM(Ho et al., 2020), DDIM(Song et al., 2020)) denoise images directly on the pixel grid, typically requiring hundreds of reverse steps and substantial GPU resources to produce high-quality images. Rombach et al(Rombach et al., 2022b). introduced the Latent Diffusion Model (LDM), which shifts the stochastic diffusion process into the latent space of a pretrained auto-encoder. This reduces dimensionality by roughly 16× while preserving perceptual details, enabling megapixel-scale synthesis on commodity GPUs.

**Formulation in Latent Space.** Let $E$ and $D$ denote the encoder and decoder of the auto-encoder, respectively.For a given input image $x$, its latent representation is obtained as $z_0 = E(x)$. The forward (noise-adding) diffusion process is defined by

$$q(z_t \mid z_{t-1}) = \mathcal{N}(z_t ; \sqrt{1 - \beta_t}\, z_{t-1},\, \beta_t\, \mathbf{I}),$$

and has the closed form

$$q(z_t \mid z_0) = \mathcal{N}(z_t ; \sqrt{\bar{\alpha}_t}\, z_0,\, (1 - \bar{\alpha}_t)\, \mathbf{I}), \qquad \text{where} \ \ \bar{\alpha}_t = \prod_{s=1}^{t}(1 - \beta_s).$$

The reverse diffusion is learned by a UNet $\epsilon_\theta$ that predicts the added noise. Using the deterministic DDIM sampler, one denoising step is

$$z_{t-1} = \sqrt{\frac{\bar{\alpha}_{t-1}}{\bar{\alpha}_t}}\, z_t + \sqrt{\bar{\alpha}_{t-1}}\left(\sqrt{\frac{1}{\bar{\alpha}_{t-1}} - 1} - \sqrt{\frac{1}{\bar{\alpha}_t} - 1}\right) \epsilon_\theta(z_t, t, c),$$

where $c$ is an optional conditioning vector. Modeling the diffusion process in latent space retains the mathematical elegance of diffusion models while drastically reducing memory and computation, which is critical for image-wise optimization in privacy-protection pipelines.

**Inversion and Editable Latent Representations.** In editable latent modeling for real photographs, we first apply deterministic DDIM inversion to map the input image $x$ into a noise trajectory $\{z_t\}$. Once this latent representation is obtained, two editing routes are available: (i) directly manipulating the noise codes—the approach adopted in this work—which is computationally lightweight and better preserves the image's overall structure; and (ii) adjusting the representation via cross-attention or embedding vectors (e.g., Null-text Inversion(Mokady et al., 2023)). Both routes support gradient-based objective optimization, enabling tasks such as identity transfer, watermark removal, or text insertion. The edited latent is subsequently decoded back into pixel space by the decoder $D$.

### A.3 NEGATIVE-PROMPT INVERSION

In null-text inversion(Mokady et al., 2023) (NTI), multiple gradient-descent updates of the unconditional embedding $\varnothing_t$ must be carried out at every diffusion step to achieve faithful reconstruction

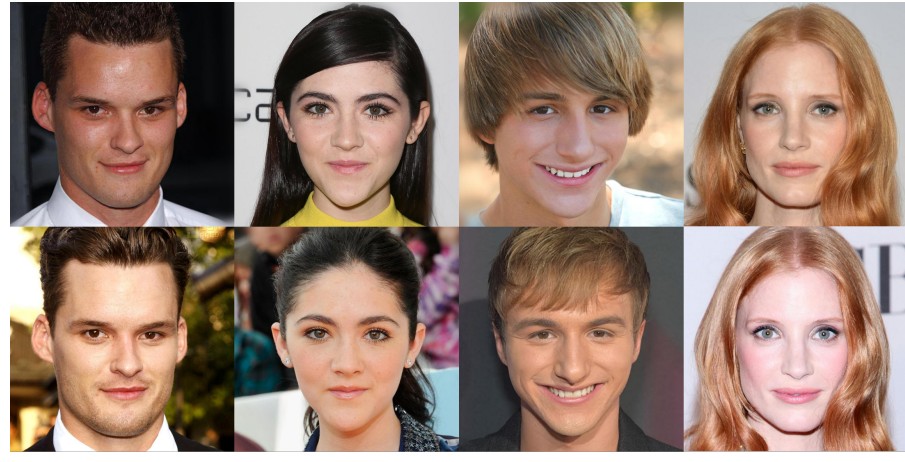

Figure 6: Target identities used for impersonation. The first row contains images used for training, while the second includes images used for testing

under classifier-free guidance (CFG), making NTI the principal computational bottleneck. Negative-prompt Inversion(Miyake et al., 2025) (NPI) eliminates this overhead by directly substituting the conditional prompt embedding $C$ for the negative branch's $\varnothing_t$, so that the positive and negative prediction paths in the CFG formulation share an identical embedding; consequently, the whole inversion is completed with a single forward DDIM(Song et al., 2020) pass and requires no backward optimization. Theoretical analysis shows that once NTI converges, the reconstruction error tends to zero when the two velocity fields coincide; under this condition the optimal $\varnothing_t$ gradually aligns with $C$ in semantic space, implying that $C \approx \varnothing_t$ yields the same reconstruction fidelity without step-wise refinement. Empirical evidence supports(Miyake et al., 2025) this approximation: at a resolution of $512 \times 512$ with 50 sampling steps, NPI finishes one inversion in about 5 second, whereas NTI still needs over 130 second, delivering a $30\times$ speed-up while maintaining virtually identical PSNR(Wang et al., 2004), LPIPS(Zhang et al., 2018), and CLIP(Hessel et al., 2022) scores. Therefore, NPI preserves high fidelity yet drastically reduces computational and memory costs, making it an ideal replacement for NTI within our two-stage attack–restoration framework.

### A.4 TARGET IMAGES

Our proposed model aims to generate protected face images that cause malicious face recognition (FR) systems to misidentify them as a specified target identity. Figure 6 shows the four target identities provided by AMT-GAN(Hu et al., 2022), as referenced in the Experiments (Section 4(**?**) of the main paper. To better approximate real-world scenarios, we ensure that the target images used for training are distinct from those used for testing.

### A.5 FACE RECOGNITION MODELS

For a fair comparison, we used publicly available pretrained face recognition (FR) models. Three of these models are based on the state-of-the-art ArcFace algorithm, which processes face images at a resolution of 112×112 and encodes them into 512-dimensional feature vectors. These three models differ in network architecture and training data: IR152(He et al., 2016) uses ResNet-152, IRSE50(Hu et al., 2018) uses ResNet-50, and MobileFace(Chen et al., 2018) is built on Mobile-FaceNet. In contrast, Facenet(Schroff et al., 2015) adopts the Inception-ResNet architecture and follows its original training protocol with an input resolution of 160×160. To evaluate performance, we report recognition accuracy on the CelebA-HQ(Karras et al., 2017) dataset: 90.70% for IR152(He et al., 2016), 90.80% for IRSE50(Hu et al., 2018), 83.00% for MobileFace(Chen et al., 2018), and 91.20% for Facenet(Schroff et al., 2015).

## A.6 PARAMETER SETTINGS

**Effect of Balance Coefficient $\lambda$.** Progressively decreasing the balance coefficient $\lambda$ in the Identity Diversion Stage loss shows(As shown in Tab.4) that moderately blending the angular-margin term markedly boosts deception: when $\lambda = 0.75$–$0.5$, the PSR jumps from $79.17\%$ to roughly $87\%$. However, further reducing $\lambda$ to $0.25$ or $0$ introduces an overly strong margin that destabilizes optimization, causing the PSR to fall to $59\%$ and $21\%$, respectively. Meanwhile, the visual-quality metrics—FID(Heusel et al., 2017), PSNR, and SSIM(Wang et al., 2004)—vary inversely: a pure cosine constraint ($\lambda = 1$) yields the best FID(Heusel et al., 2017), whereas dominance of the angular-margin term significantly degrades naturalness. Overall, setting $\lambda$ in the range $0.5$–$0.75$ achieves the best trade-off between deception strength and visual quality, while relying exclusively on either constraint breaks this balance.

| $\lambda$ | PSR ($\uparrow$) | FID ($\downarrow$) | PSNR ($\uparrow$) | SSIM ($\uparrow$) |
|---|---|---|---|---|
| 1 | 79.17 | **18.0380** | **27.8664** | **0.8538** |
| 0.75 | 86.14 | 31.3818 | 25.5461 | 0.8052 |
| 0.5 | **87.61** | 31.8926 | 25.3224 | 0.8022 |
| 0.25 | 59.00 | 30.9101 | 25.4267 | 0.8026 |
| 0 | 20.63 | 28.3098 | 25.6824 | 0.8077 |

Table 4: Effect of Balance Coefficient $\lambda$

**Effect of Step Count in Identity Diversion.** In the step-count ablation for the Identity Diversion stage (Tab.5), we increase the optimization iterations from 30 to 35 and 40 while tracking PSR, FID, PSNR, and SSIM. As the step count rises, deception strength improves consistently: PSR increases from $84.32\%$ to $87.61\%$ and then to $88.85\%$. However, the 40-step setting noticeably raises per-image inference time and prolongs the overall training schedule relative to 35 steps. Balancing accuracy and efficiency, 35 steps achieve nearly the same PSR as 40 steps while substantially reducing runtime and compute, making 35 steps the more cost-effective choice for practical deployment.

| Step | PSR ($\uparrow$) | FID ($\downarrow$) | PSNR ($\uparrow$) | SSIM ($\uparrow$) |
|---|---|---|---|---|
| 30 | 84.32 | 29.6444 | 25.8829 | 0.8123 |
| 35 | 87.61 | 31.8926 | 25.3224 | 0.8022 |
| 40 | 88.85 | 34.8996 | 24.8544 | 0.7859 |

Table 5: Effect of Step Count in Identity Diversion.

**Effect of Step Count in Visual Fidelity Restoration.** In the step-count ablation for Visual Fidelity Restoration, we evaluated 0, 50, 100, 150, and 200 iterations. The jump from 0 to 50 steps yields the most pronounced quality gain: FID(Heusel et al., 2017) drops from 31.89 to 17.05, while PSNR and SSIM(Wang et al., 2004) improve in tandem. Increasing to 100 steps brings further, but clearly diminishing, returns. Beyond 100 (i.e., at 150 and 200 steps), the metrics nearly plateau, whereas inference and training time grow almost linearly. Following these trends, we adopt a 30–70 step window: this range already delivers strong image quality while keeping computational overhead moderate, offering a favorable balance between visual fidelity and efficiency.

| Step | FID ($\downarrow$) | PSNR ($\uparrow$) | SSIM ($\uparrow$) |
|---|---|---|---|
| 0 | 31.8926 | 25.3224 | 0.8022 |
| 50 | 17.0469 | 26.8686 | 0.8424 |
| 100 | 15.8143 | 27.1990 | 0.8509 |
| 150 | 15.5435 | 27.2784 | 0.8528 |
| 200 | 15.4212 | 27.3156 | 0.8540 |

Table 6: Effect of Step Count in Visual Fidelity Restoration.

## A.7 MORE VISUALIZATION RESULTS

This section presents additional visualizations to validate the proposed two-stage framework. The first row shows the original images; the second row displays the results produced by the Identity Diversion stage; and the third row presents the outputs of the Visual Fidelity Restoration stage. As observed, the second row achieves a substantial increase in PSR but at the cost of some degradation in visual quality; by contrast, the third row largely preserves PSR while effectively restoring naturalness and fine details, resulting in superior overall perceptual quality.

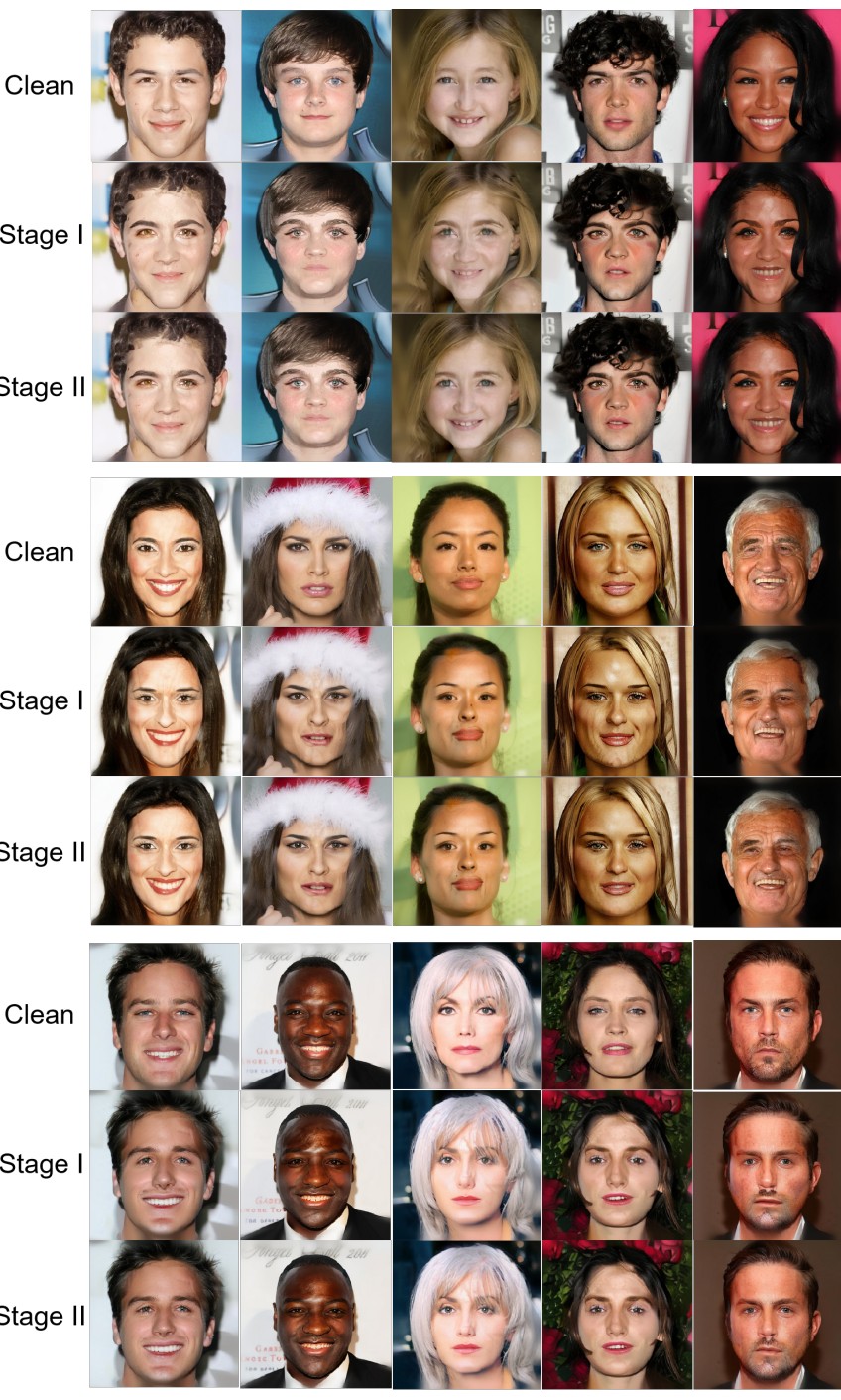

Figure 7: More Visualization Results

