# OpenReview forum: "From Attack to Restoration: A Two-Stage Diffusion Framework for Face Privacy"
_ICLR.cc/2026/Conference — ICLR 2026 Conference Withdrawn Submission_

### Official Review · Reviewer_jrx1 · 2025-10-14

**Soundness:** 2
**Presentation:** 2
**Contribution:** 2
**Rating:** 2
**Confidence:** 5

**Summary:**

The paper proposes a two-stage diffusion-based framework for facial privacy protection. In the first stage (Identity Diversion), the authors introduce Negative Prompt Inversion (NPI). In the second stage (Visual Fidelity Restoration), the method employs perceptual and regularization losses to enhance visual realism. Experimental results on CelebA-HQ and LADN suggest improvements in protection success rate and visual quality compared to prior work.

However, the paper suffers from fundamental flaws in its privacy protection design, unclear presentation, weak originality, and unsatisfactory qualitative results. Therefore, I recommend rejection.

**Strengths:**

- The experiments were extensive.
- The research problem is very meaningful.

**Weaknesses:**

**1. Fundamentally flawed privacy protection mechanism.**
- The proposed two-stage framework appears reasonable at first glance, **but is conceptually incorrect**. In the second stage, the Restoration module takes the original face as input, which inherently contains the subject’s true identity (Authors can explore the concept of differential privacy). This design introduces a direct privacy leakage risk and contradicts the goal of privacy protection.
- Furthermore, if the two-stage framework is intended merely to stabilize multi-objective training, its contribution is marginal. There exist numerous well-established methods to achieve stable optimization, and the proposed design does not represent a superior or novel solution.

**2. Poor presentation and unclear results.** It is unclear where the scores displayed below the original faces originate from, and I could not find any explanation for Figure 1 in the main text. The lack of clarity in figures and captions significantly hinders the reader’s understanding of the results.

**3. Questionable contributions.** It is unclear why the first and second listed “contributions” are considered novel. The so-called NPI component appears to be based on previously published techniques, and thus does not constitute a new or significant contribution.

**4. Weak visual results.**  The qualitative results in Figure 7 are weak. The first male face becomes noticeably feminized after protection, while the second male face also shows substantial geometric distortion, particularly around the mouth. Overall, the visual quality is unsatisfactory. The presented approach resembles face swapping rather than a genuine adversarial protection method.

**Questions:**

**1. Privacy Mechanism  Justification..** In the proposed two-stage framework, the second-stage Restoration module takes the original face as input, which inherently contains the subject’s true identity. How does this design prevent direct identity leakage?

**2. Novelty and contribution justification.** The paper lists several contributions, but it remains unclear which aspects are genuinely novel. Could the authors explicitly distinguish their proposed components from prior techniques?  If the second stage primarily aims to stabilize multi-objective optimization, how does it differ from existing optimization stabilization techniques? What makes it a distinct contribution rather than a training trick?

**3. Visual results justification.**  To what extent does the proposed approach differ from face-swapping–like transformations, and how do the authors ensure it aligns with adversarial protection objectives?



The author should be reminded that they must provide sufficiently compelling reasons to refute my viewpoint; otherwise, I will continue to reject this work in ICLR.

---

> ### Author Response · Authors · 2025-11-30
>
> Thank you very much for your valuable feedback; these suggestions will serve as the core direction for me to refine this paper moving forward.

---

### Official Review · Reviewer_Xk7L · 2025-10-29

**Soundness:** 2
**Presentation:** 2
**Contribution:** 2
**Rating:** 2
**Confidence:** 4

**Summary:**

This paper studies the problem of anti-facial recognition by developing a two-stage diffusion-based framework for crafting adversarial yet imperceptible perturbations. In the first stage, the framework proposes using negative prompt inversion (NPI) during the reverse diffusion process, combined with an angular-margin constraint, to push the protected face’s features toward a target identity. In the second stage, the method refines the output to achieve better visual quality while maintaining the adversarial effect. Extensive experiments demonstrate the advantages of the proposed framework.

**Strengths:**

+ The paper focuses on a timely and important problem of protecting facial privacy against unauthorized facial recognition models.

+ The use of diffusion models for privacy is an emerging and interesting research direction.

+ The experiments show the proposed method achieves state-of-the-art performance in both visual quality and protection success (under impersonation settings).

**Weaknesses:**

- No evaluations are conducted under obfuscation/dodging scenarios, leaving a gap between the actual privacy goal in many use cases

- The proposed framework is complex and difficult to comprehend. No algorithm pseudocode provided. Some implementation details and design insights are vaguely described

- No evaluations or discussions in terms of the robustness of the method with respect to image transformations (e.g., compression, purification)

**Questions:**

While the paper tackles an important and timely research topic of anti-facial recognition, the paper falls short in the following critical aspects:

1. The proposed method is purely designed to impersonate a given face image of a target identity (different from the victim). It is unclear whether the proposed method is effective in more privacy-relevant obfuscation scenarios (a.k.a. dodging). Note that successful impersonation does not necessarily imply successful dodging. Prior literature has already highlighted the differences between the two settings (see [1] and Section 5 of [2]), which should be taken into consideration in the experiments and/or discussions.

    [1] Rethinking Impersonation and Dodging Attacks on Face Recognition Systems, https://arxiv.org/pdf/2401.08903v4

    [2] Enhancing Facial Privacy Protection via Weakening Diffusion Purification, https://arxiv.org/pdf/2503.10350

2. The proposed framework is quite complex, including two stages and multiple components per stage. The insights into why these stages/components are important/necessary are not explained well in Section 3. In Stage 1 (Identity Diversion), three loss functions are introduced: $L_{Cosine}$, $L_{Arc}$, and $L_{attn}$. Are they all necessary to achieve the goal of identity diversion? Their motivations are not well elaborated in Section 3.3, making it difficult for readers to appreciate their importance. Additionally, the role of the second stage is argued to be further improving the perceptual quality. Do we really need to separate the framework into two stages? It appears that separating into two stages may double the computational costs and result in unfair comparisons with existing methods, as they do not involve an additional step for further refinement. That being said, the authors are recommended to discuss the computational costs of their method, with comparisons to prior competitive alternatives such as WDP and DiffAM.

3. The paper does not provide pseudocode for the algorithm, making it difficult to understand its exact implementation. For example, the method proposes replacing null-text embedding with negative prompt inversion (NPI); however, the implementation details of NPI are unclear. How is the negative prompt inverted? How are the hyperparameters set? Do you make any thoughtful adaptations to improve the performance, or is this a straightforward application of an existing method? The lack of clarity of the proposed framework leaves some doubt about whether the experimental results can be reproduced.

4. The paper does not provide evaluations or discussions on how robust the method is with respect to image transformations, such as image cropping, JPEG compression, Gaussian corruptions, simple denosing operations, etc. As mentioned in the abstract, the work is motivated by the observation that diffusion-based privacy methods are typically vulnerable to purification effects. However, it is somewhat surprising to me that no experiments are provided to evaluate whether the proposed framework is more resilient to the purification effect.

5. The quality of the writing needs to be improved, especially for the introduction. There should be a blank space between the citation and the preceding word. When referring to multiple references, it is typical to place multiple citations inside a single bracket. In addition, lines 089 and 140 have obvious spacing issues. The authors should do a better job in proofreading before submission, and are recommended to review the accepted papers from previous years to familiarize themselves with the typical ICLR citation format.

6. The proposed method can be potentially misused by adversaries for malicious purposes. The paper does not provide a statement to clarify such an ethical concern.

**Details Of Ethics Concerns:**

The proposed method achieves an impressively high impersonation success rate for a given target identity, which may be misused by adversaries for malicious purposes. The paper does not provide any statement to discuss the potential ethical concerns.

---

> ### Author Response · Authors · 2025-11-30
>
> Thank you very much for your valuable feedback; these suggestions will serve as the core direction for me to refine this paper moving forward.

---

### Official Review · Reviewer_RAPw · 2025-10-30

**Soundness:** 2
**Presentation:** 3
**Contribution:** 2
**Rating:** 4
**Confidence:** 5

**Summary:**

This paper presents a two-stage diffusion framework for facial privacy protection. The first stage, Identity Diversion, uses Negative Prompt Inversion and an angular-margin loss to mislead face recognition models, while the second stage, Visual Fidelity Restoration, refines the image to restore visual quality.

**Strengths:**

1. The two-stage strategy is intuitive yet practically useful. NPI stabilizes inversion and improves convergence; the restoration stage successfully recovers natural appearance without losing deception strength.
2. The pipeline is clearly described and supported by diagrams and quantitative results.

**Weaknesses:**

1. The attack is tested on a small number of closed-set FR models. It is unclear whether the protection generalizes to unseen datasets or commercial FR systems.
2. There is no study on the angular-margin strength or alternative margins (sensitivity) (correctness/clarity). No direct evidence found in the manuscript.

**Questions:**

Please refer to the weaknesses section.
In addition, could you explain why the dynamic rollback mechanism helps stability or convergence?

---

> ### Author Response · Authors · 2025-11-30
>
> Thank you very much for your valuable feedback; these suggestions will serve as the core direction for me to refine this paper moving forward.

---

### Note · Authors · 2025-11-30

I have read and agree with the venue's withdrawal policy on behalf of myself and my co-authors.